# A Review of Recent Developments in Veterinary Otology

**DOI:** 10.3390/vetsci9040161

**Published:** 2022-03-25

**Authors:** Richard Harvey

**Affiliations:** Willows Referral Service, Shirley, Solihull B90 4NH, UK; richard.harvey@willows.uk.net

**Keywords:** dog, canine, otitis, ear canal, diagnostic, therapy, review

## Abstract

The knowledge gap between practical research and its implementation in veterinary practice is becoming harder to bridge, as researchers now have a plethora of journals in which to publish. This paper summarizes recent research from the latest publications related to ear disease in dogs which have implications for veterinary practitioners. The topics reviewed include 16s rRNA new-generation sequencing, the use of oclacitinib in pinnal ulceration, the etiopathogenesis of aural hematoma, contamination of the middle ear during elective myringotomy and how to avoid it, and the use of carbon dioxide lasers in chronic obstructive otitis.

## 1. Introduction

Veterinary regulators require us to keep up to date, and our professional pride makes us want to keep abreast of new knowledge. However, keeping up with advances in veterinary knowledge is no longer as simple as reading your monthly professional journal. More and more publishing opportunities are available for authors to consider, meaning there are more and more journals for practitioners to read. Clinicians must find time to search sites such as cabi.org and pubmed.ncbi.nlm.nih.gov on a regular basis.

This review aims to bring veterinary practitioners up to date with recent publications relating to canine ear disease.

## 2. New Technology and Otic Microflora

This very large study [1] comprised 257 samples from 128 healthy ears and 129 ears with otitis. The objective was to compare the bacterial and fungal microbiome between the two groups, utilising next-generation sequencing targeting 16s rDNA as well as qPCR primers targeting the ribosomal 16s. Shotgun sequencing was used to quantify the abundance of organisms and to confirm the identity of organisms.

The top three most dominant bacteria, based on average relative abundances, in healthy ear samples were *Cutibacterium acnes* (previously known as *Propionibacterium acnes*), *Staphylococcus pseudintermedius*, and *Streptococcus* spp. In general, there was more diversity in normal ears. In some instances, some 30 different species of bacteria were identified from a normal external ear canal (Figure 1). None of the samples from healthy ears yielded zero isolates.

This overall picture of a diverse flora in the normal external ear canal, agrees with the more familiar techniques of swab and culture methodology, notwithstanding that culture frequently fails to identify the vast majority of species that 16s rDNA next-generation sequencing can demonstrate. However, swab and culture—not infrequently—yield zero growth, from both healthy and apparently infected ears (personal observation).

In this study, the samples from affected ears of dogs with otitis externa were compared to healthy samples. In total, 78.3% of the clinically affected ear samples had microbial overgrowth; 62.78% bacterial overgrowth, 8.5% fungal overgrowth, and 7.0% had both bacterial and fungal overgrowth. The most abundant microbial taxa in clinically affected ears were *Malassezia pachydermatis*, *S. pseudintermedius*, *S. schleiferi*. In addition, there were a few anaerobic bacteria such as *Finegoldia magna*. Approximately 21% (27 of the 129 samples) of the clinically affected ear samples were observed to be dominated by a single species, either bacterial or fungal, that constituted more than 90% of the relative abundance in the sample. None of the samples from affected ears yielded zero isolates.

Again, the reduction in overall diversity is recognised in swab and culture methodology but, again, the vast majority of species demonstrated by 16s rDNA technology were not identified by traditional culture methods.

The question for the clinician is this: other than identifying that the otic biome is far more complex than we thought, does it tell us anything useful *vis a vis* dealing with otitis? The identification of a potential opportunistic pathogen, *F. magna*, is interesting. *F. magna* is a Gram-positive anaerobic coccus, which is not generally considered pathogenic in canine ear and skin infections. However, in man, *F. magna* has been recently recognized as an opportunistic pathogen [2]. While *F magna* could be found in normal ears, Tang et al. [1] found it to be much more abundant (9.1 × 10^7^ 16S copies/sample) in affected ears. *F. magna* can elaborate neutrophil activation toxins [3], can penetrate human epidermis [4] and, in association with *Pseudomonas aeruginosa*, can be found within joint biofilm infections in human leg ulcers [5].

*F. magna* is extremely fastidious in its growth requirement, which it is why its importance may be under-recognised in canine practice.

## 3. Oclacitinib for Pinnal Tip Ulceration

Ear tip ulcerative dermatitis (ETUD) is a clinical reaction pattern, suggestive of vascular damage.

Histopathological changes may be inflammatory or noninflammatory, and may be associated with thrombus formation in some cases. Differential diagnosis includes stable fly bites [6], leishmaniosis [7], bartonellosis (*Bartonella henselae*), experimental infection by *Rickettsia rickettsia* [8], various frostbites, cryoglobulinaemias and cryofibrinogenaemias, and breed-specific (familial) vasculitides [9]. The final differential is proliferative thrombovascular necrosis of the pinnae [10].

Ear tip ulcerative dermatitis (Figure 2) is usually an idiopathic diagnosis [11] characterised by histological features of vascular damage to small dermal arterioles, with or without fibrin thrombi. In most cases, histopathological or immunopathological demonstration of vascular damage is lacking [11]. 

The authors of this paper report the successful use of oclactinib in 25 cases of ETUD [11]. 

Oclacitinib is a first-generation Janus kinase (JAK) inhibitor, approved in many countries to treat atopic dermatitis, typically at a dose of 0.4 to 0.6 mg/kg. At higher, extralabel doses, oclacitinib has immunosuppressive activity in vitro, with inhibition of T-cell proliferation and secretion of activator cytokines IL-2, IL-15, proinflammatory cytokines interferon-gamma, IL-18 and the regulatory cytokine IL-10 [12].

The aims of this retrospective study [11] were to describe the clinical features of 25 cases of idiopathic, chronic ETUD, and to report response to treatment with oclacitinib. In total, 22 cases had complete resolution, within 1–3 months. Four cases relapsed within one month when treatment was withdrawn and one dog required long term therapy to maintain remission. Given that oclacitinib is immunosuppressive it would be prudent to rule out infectious diseases before initiating treatment.

This report is important because one of the treatment modalities available is surgical amputation of the pinnal tip. Patently, an effective medical treatment is a very useful alternative.

## 4. New Proposals on the Aetiology of Aural Haematoma

Aural haematoma is common (Figure 3), accounting for 0.25% of cases in a very large survey of the medical records of 905,554 dogs in 2016 from 887 veterinary practices in England [13]. Proposed aetiologies have included excessive head shaking [14], and an unidentified immunological process [15,16]. 

The huge study [13] reports on 2249 cases of aural haematoma in this overall population of 905,554 dogs. A general increasing risk with age of dog and increasing adult body weight was noted, and a breed predisposition identified in French bulldog, English and Staffordshire Bull terriers, Golden retrievers and St Bernard. In addition, a folded or semi-erect carriage was found to predispose (Figure 4). The authors note that dogs with erect pinnae have no pinnal fold, while those with pendulous ears have a degree of rigidity at the base, where the fold occurs.

Increasing age has been shown to affect cartilage in man [17]. Human auricular cartilage shows increased reorganisation and heterogeneity of elastic fibre diameter with advancing age, suggesting reducing elasticity over time. The authors of this paper postulate that these finding could mirror changes in dogs [13]. An age-related loss of elasticity might lower the threshold for mechanical failure in the pinnal cartilage, at the level of the fold. 

There is a degree of rigidity at the base of the pinna and a fold, around mid-pinna, might be subject to repeated flexing, for example in a dog with otitis externa or damage to the distal pinna. This might result in trauma to the cartilage, and aural haematoma.

This is an important publication which allows us to inform our clients with much more confidence about this relatively common condition, which often requires surgical reduction, under general anaesthesia.

## 5. Myringotomy

Elective myringotomy in the dog is used to gain access to the middle ear, to allow sampling for bacteriology, in particular, or cytology. This clever cadaver study [18] used standard techniques to flush and clean an external ear canal, then one drop fluorescein in 5 mL water was inserted and residual fluid removed by suction. This left a thin film of fluorescein on and around the tympanic membrane (Figure 5).

Next, a standard, elective myringotomy was performed with a myringotomy needle passed through the working channel of a veterinary otoscope. A catheter was passed into the middle ear and any fluid removed. If only air was aspirated then 1 mL sterile saline was instilled, and then aspirated. The colour of the fluid obtained from the middle ear was judged as either yellow or colourless [18].

In 19 of 28 ears, 67.9%, fluorescein was found in the middle ear samples, indicating contamination from the external ear canal using standard procedures.

Having identified the problem, the Vienna group found a solution. The group designed a special table, such that the ear to be investigated could be accessed from under the dog that was positioned with affected ear downwards. The fluorescein study of Reinbacher et al. [18] was repeated but this next study exploited ventral access to the tympanic membrane [19]. This resulted in a contamination rate of 2 out of 36 (5.55%) [19]. 

The clinical importance of this study is hard to exaggerate but it is difficult to see how practitioners can perform a ventral approach in this manner without substantial investment, even assuming that they have the space available. At the very least, a premyringotomy clean-and-dry cycle should be followed by a swab for culture and sensitivity taken immediately adjacent the tympanum, such that this sample can be compared to that taken from the middle ear, microbiota in the middle ear and the external ear canal frequently differ from each other [20].

## 6. Laser Ablation of Chronic Hyperplastic Obstructive Otic Tissue

This report describes carbon dioxide laser surgery for the management of obstructive and proliferative otitis in 26 dogs [21]. In total, 42 ears with chronic, proliferative hyperplastic otitis and hyperplastic stenotic otitis underwent laser surgery. The laser was used to ablate the hyperplastic tissue from both the entrance to the ear canal and the vertical ear canal (Figure 6). 

Of great interest was the ability of the laser to reach the junction of the vertical and horizontal ear canal, to ablate stenotic hyperplastic tissue. Without such laser surgery, a lateral wall resection would be required to allow access for traditional surgery. 

Other advantages over “cold steel” surgery include the ability to sculpt rather than just excise, sterilisation of the wound bed, and effective haemostasis. In addition, the laser ablates sensory nerve endings, helping to reduce postoperative pain. As a consequence, most of the dogs required no postoperative pain relief. 

Postoperative treatment included oral ciclosporin, and pain relief if required. Most dogs received topical application of silver sulfadiazine cream once or twice daily until the wounds had healed. In 21 dogs a single episode of laser treatment was sufficient, 3 dogs required a second laser treatment. Nine dogs had antimicrobial-saturated ear wicks placed.

Excised tissue from 17 ears was submitted for histopathological examination. Eight ears had a predominantly sebaceous gland hyperplasia, three had a predominantly ceruminous response and six had a mixed glandular pattern. Epidermal hyperplasia, inflammation and fibrosis varied from mild to severe.

This study is relevant as many institutions now offer laser surgery. Nonlaser options include surgery, ciclosporin [22], and multiple, intralesional, injections of small amounts of potent steroid, such as triamcinolone [23], and these are much less successful. We await further studies, perhaps comparing CO2 lasers with cold steel, or even a different laser to which the outcome could be compared.

## 7. Visualising Biofilm in Canine Otitis Cytology Smears

Micro-organisms, particularly (but not exclusively) *Pseudomonas aeruginosa*, associated with canine otitis externa may cause biofilm-associated infections. A key component of biofilm is microbial aggregate and extracellular polymeric substance. Identifying aggregate-associated infection (AAI) biofilm from ear samples in-clinic is very difficult.

This study [24] set out to establish whether periodic-acid Schiff (PAS) or a modified Wrights stain (such as Diff Quik) can stain clinical otic cytological smears adequately to identify the polysaccharide matrix on cytological smears. In addition, the reproducibility of identification of microbial aggregates within a discrete area of stained matrix was assessed. A total of 40 privately-owned dogs yielded 40 pair samples. The investigators concluded that PAS staining was superior to modified Wrights stain [24].

Key findings *vis a vis* identifying potential AAI were the presence of three or more aggregates, the presence of high-density material, extracellular staining and the presence of discretely stained matrix. 

The extracellular polymeric substance is composed mainly of polysaccharides, proteins, nucleic acids, and lipids; they provide the mechanical stability of biofilms, mediate their adhesion to surfaces and form a cohesive, three-dimensional polymer network that interconnects and transiently immobilizes biofilm cells [25]. The bacteria are within this matrix, Figure 7, and since it is three-dimensional, the bacteria are at different levels within. Thus, the observation that the bacteria are in different focal plains within these aggregates of stained matrix is highly suggestive of biofilm.

The authors stress that this technique has yet to be verified by “gold standard” techniques such as confocal laser scanning microscopy or fluorescence in-situ hybridisation. The aim is to document an in-house light microscopic that can support a diagnosis of biofilm-associated infection.

## 8. Collaborative Care Improves Patient Outcome

This retrospective study tries to quantitatively assess the improved patient care that follows after a primary care veterinarian refers a case of chronic or recurrent otitis externa to a Board-Certified veterinary dermatologist [26]. Medical records of 65 client-owned dogs were reviewed. The average duration of otitis prereferral was 10 months. Cocker Spaniels and Labrador retrievers were the most commonly seen breeds affected. The study concluded that referral results in some quantifiable benefits:the number of episodes of otitis was reduced;median time between recurrence of an episode of otitis was reduced;proliferative pathological change was significantly reduced;dogs with chronic otitis had a better outcome if referral occurred within 6 months of chronic otitis treatment.

These results might reflect the increased awareness of the primary, predisposing, and perpetuating causes that specialist knowledge and practice can bring, allowing a comprehensive “staging” of the case. Of the 65 cases which were the basis for the study, 93.8% were diagnosed with atopy as the underlying, predisposing, cause. 

This failure to arrive at the correct diagnosis, particularly in the case of recurrent otitis externa, likely plays a part in treatment failures and recurrent episodes of disease, and so frustrations occur. The authors make the point that the paper did not study barriers, if any, either to investigatory procedures or early referral by the referring veterinarian.

Other factors which might aid the referral veterinarian’s ability to positively impact on the cases of chronic or recurrent otitis externa could be that referral facilities:have the equipment to perform effective, deep cleaning of the external ear canal;tend to have access to advanced imaging modalities, such as video-otoscopy and CT scanning;board-certified veterinary dermatologists in referral institutions can call upon colleagues in associated fields such as neurology and soft tissue surgery, for specialist input.

All of these factors can positively affect patient outcome, as this paper shows [26].

## 9. *Malassezia pachydermatis* Otitis Unresponsive to Primary Care: Outcome in 59 Dogs

This retrospective paper [27] describes the specific management of unresponsive otitis externa in 50 dogs referred to the Queen Mother Hospital for Animals at the Royal Veterinary College over a period from 2007 to 2018. The group originally comprised 59 dogs, but 9 were excluded as they had malassezial otitis media in addition to malassezial oitis externa. This final group of 50 dogs had a final diagnosis of malassezial otitis externa, either unilateral (36 dogs) or bilateral (23 dogs). 

*Malassezia pachydermatis* (Figure 8) is recognised both as a commensal and an opportunistic pathogen in the dog’s ear [28]. Many cases, most often found as mixed bacterial and malassezial infections, respond to commercially available polypharmaceutical products containing a mix of antibacterial agents, antifungal agents, and glucocorticoids [29]. However, a proportion, estimated as between 8 and 26%, [30,31], have *Malassezia pachydermatis* cited as the primary pathogen.

The initial consultation and diagnosis having been made, an appointment was made for admission, general anaesthesia, video-otoscopic examination, ear flush, and CT scan. The dermatological unit at the Royal Veterinary College routinely prescribe a course of anti-inflammatory treatment for the (typically) 2-week period between this first appointment and the day of admission. The dogs were administered anti-inflammatory doses of prednisolone, with an average dose of 0.8 mg/kg/day or methylprednisolone, at an equivalent dose. After ear flushing, usually only one episode, dogs were prescribed a continued dose of oral glucocorticoid and, usually, a commercial polypharmaceutical otic preparation until the otitis had resolved, a median time of 27 days.

This paper demonstrates that it is not a “magic bullet” that resolved these refractory cases but a recognition of the underlying primary, predisposing and perpetuating causes. In these cases, typically underlying allergy and accumulated debris within the externa ear canal. The preadmission treatment with low dose glucocorticoid would suppress the allergic otic inflammation, and might begin to reduce any consequential oedematous and inflammatory changes within the externa ear canal. 

The other factor that helped the resolution of these chronic cases was ear flushing under general anaesthesia. This procedure removes hair, inflammatory exudate, cerumen, and any otic debris. Thorough cleaning, and drying, of the affected external ear canal facilitates examination of the tympanum and, just as important, facilitated subsequent treatment, often impeded by the material which has been flushed out [32]. The requirement for general anaesthesia is often not appreciated by owners—these procedures cannot be performed under sedation.

## 10. Conclusions

Otitis externa, otitis media, and pinnal diseases are commonly seen in small animal practice. As a consequence, there are many clinicians and researchers conducting studies and publishing papers. With scientific databases such as Pub Med being but a click away, it takes a few moments a month to keep up to date in those areas that are of interest.

## Figures and Tables

**Figure 1 vetsci-09-00161-f001:**
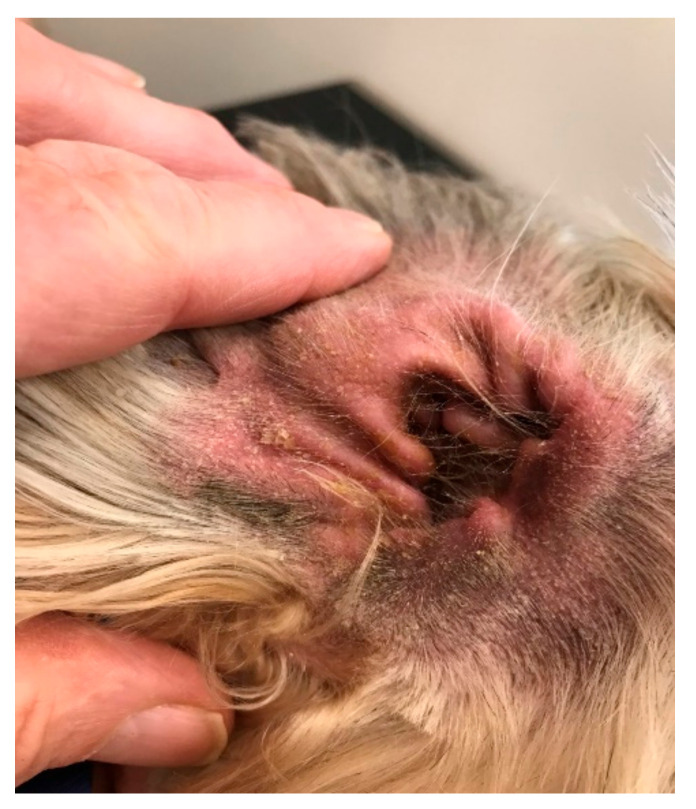
Standard swab and culture from the vertical ear canal of this case, for example, will yield perhaps four species of bacteria and one species of yeast. If the sample is subject to 16s rDNA sequencing, it will likely reveal in excess of 30 species of bacteria.

**Figure 2 vetsci-09-00161-f002:**
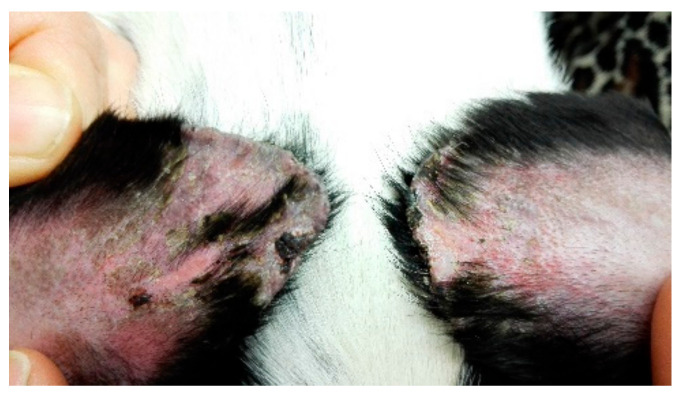
Pinnal vasculitis in a dog, before (on the left) and after treatment (on the right) Courtesy Dr. L. Cornegliani Clinica Veterinaria Città di Torino, c.so Traiano 99/d, 1-10135 Torino, Italy [11].

**Figure 3 vetsci-09-00161-f003:**
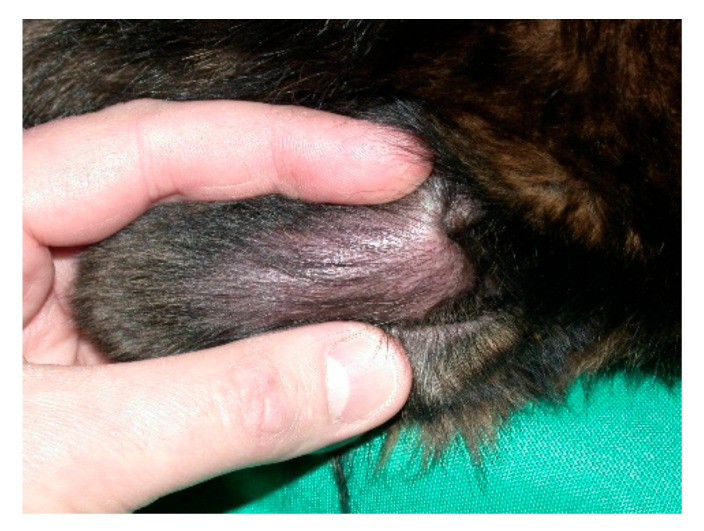
Aural haematoma in the pinna of a dog. Courtesy Dr. Stephen Baines, Willows Referral Service, Highlands Road Shirley, Solihull B90 4NH, UK.

**Figure 4 vetsci-09-00161-f004:**
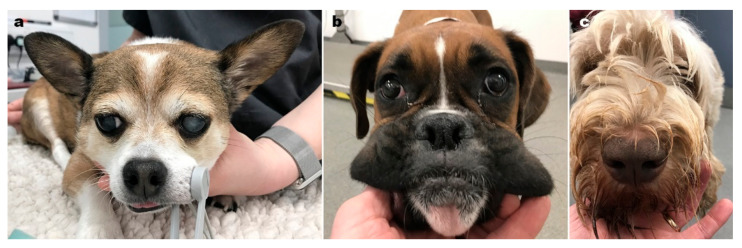
Ear carriage in dogs: (**a**) erect, (**b**) semi-erect and (**c**) pendulous ears.

**Figure 5 vetsci-09-00161-f005:**
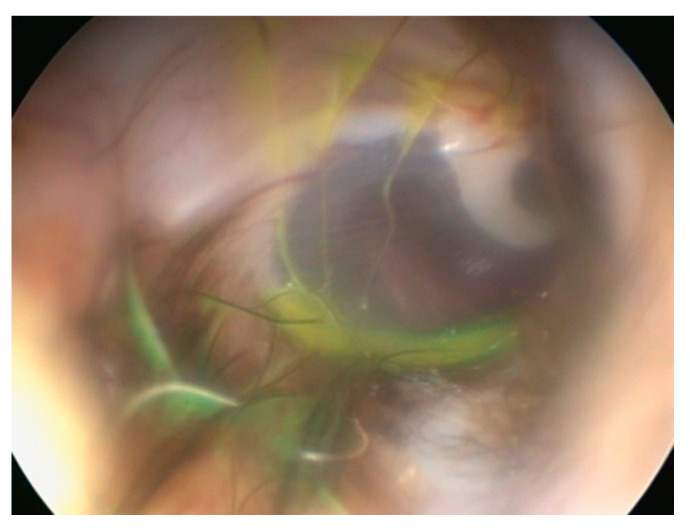
External ear canal cleaned and dried, pre myringotomy. Note that a trace of fluorescein remains. Courtesy Elisabeth Reinbacher, Clinical Unit of Internal Medicine Small Animals, Department for Companion Animals and Horses, Vetmeduni Vienna, Veterinärplatz 1, 1210 Vienna, Austria [18].

**Figure 6 vetsci-09-00161-f006:**
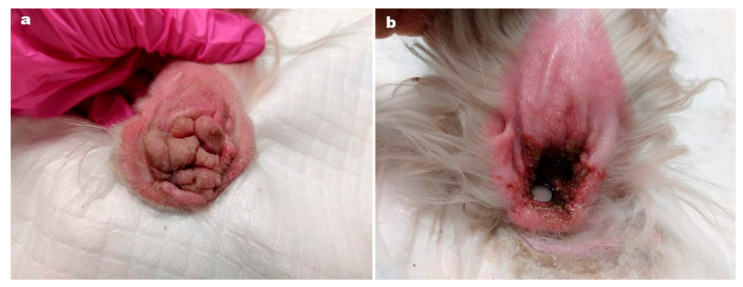
Laser surgery of chronic hyperplastic occlusive otitis, courtesy Dr. Jeylan Aslan, Dermatology for Animals, 263 Appleby Road, Stafford Heights, QLD 4053, Australia [21]. (**a**) Preoperative photo demonstrating occlusive grossly hyperplastic, (**b**) immediate postoperative picture showing removal of all excess tissue, minimal haemorrhage, and patent external ear canal.

**Figure 7 vetsci-09-00161-f007:**
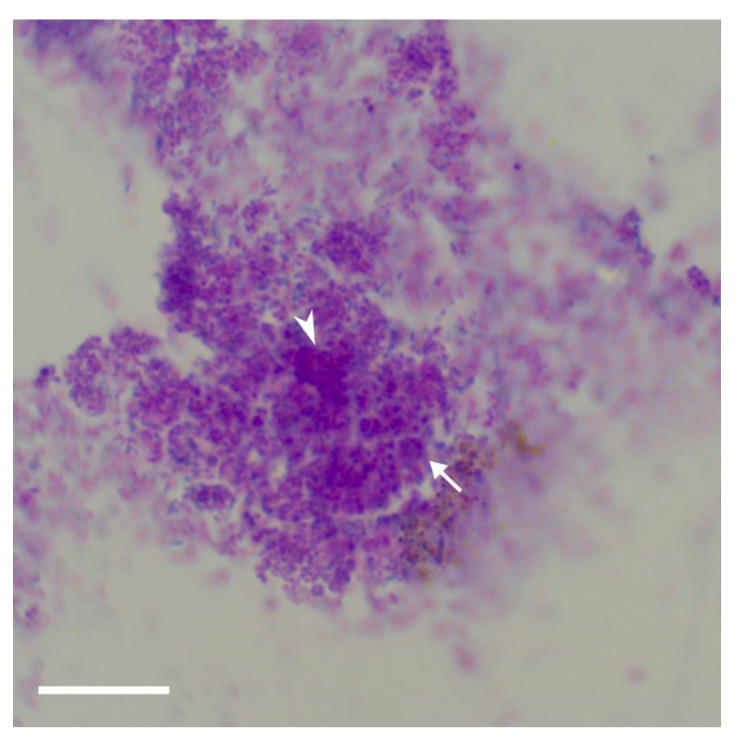
Periodic-acid Schiff stain showing bacteria within stained aggregate. As this is three-dimensional, we can see in-focus bacteria, indicated by white arrows, and out-of-focus bacteria, indicated by white arrowheads, Courtesy Henrietta Parnell-Turner, Animal Dermatology Clinic, 5610 Kearney Mesa Road, San Diego, CA 92111, USA [24]. Bar = 15 µm.

**Figure 8 vetsci-09-00161-f008:**
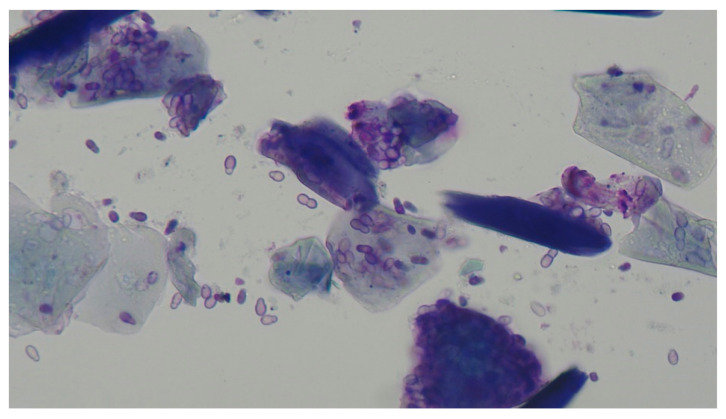
*Malassezia pachydermatis* yeast in a cytology smear from the external ear canal of a dog with malassezial otitis externa. Courtesy Prof. R. Bond, Royal Veterinary College, Hawkshead Lane, Brookmans Park, Hatfield AL9 7TA. UK.

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
