# Peer review of "A Review of Recent Developments in Veterinary Otology"

_vetsci, 2022, doi:10.3390/vetsci9040161_

Round 1

Reviewer 1 Report

Brief summary:

This review manuscript aims to provide an update on several canine otology studies and their findings and state the potential useful clinical applications to the general practitioner.

This review is relevant as general practitioners struggle with managing canine otitis and a quick read of new clinically applicable information may help strengthen their clinical acumen. Of particular importance is the review of collaborative care and the need to refer chronic otitis externa cases to provide the gold standard of care for the patient.

References are up to date and mostly within the last 5 years; although the formatting and font need to be addressed particularly from references 27-31.

The review is clear and rather concise however often times colloquial terms are used instead of scientific terms.

Title: A review of recent developments in veterinary otology

could add the word veterinary “canine” otology in the title to be more specific and possibly increase visibility using online search terms

Abstract:

Line 8: Instead of using the phrase “this paper brings together” use “this paper summarizes” recent research from …

  • The word implication is used in consecutive sentences which is generally not preferred.

Line 9: Instead of “The implications” replace with “Topics reviewed” are the use of 16s rRNA, replace the word new with “next” generation sequencing as this is the proper terminology and vets now understand the word sequencing, the use of oclacitinib for pinnal ulceration,

Line 10 replace new findings to be more specific with etiopathogenesis of aural haematomas….

Line 12 replace “is” with the plural “are” as topics is plural.

Keywords:

Line 13: why is cat a keyword? Should have canine as a keyword as well as the word dog

  1. Introduction:

Line 23: remove the word “feline” as none of the reviewed papers are related to feline otitis

  1. Line 24: Replace 16s new technology” with the exact method “16s rDNA next generation sequencing” as 16s could refer to PCR which is an old method for identification.

Line 26-27 In the Tang study next gen sequencing targeting 16s rDNA as well as qPCR primers targeting the ribosomal 16s and shot gun sequencing was used quantify the abundance of organisms and to confirm the organism identify from NGS results, respectively. Please update this section to correct technology.

Line 26: add “bacterial and fungal” before microbiomes

Line 42: remove “technology” and replace with next generation sequencing and provide a reference for this statement

Lines 44-46: The written percentages don’t total to 78.3%, this is because they are not correct as to what is presented in the Tang et al paper. Bacterial overgrowth was 62.7%, fungal overgrowth is 8.5% and 7% had both fungal and bacterial overgrowth.

Line 46: How were these taxa deemed the “most important”? These were the most abundant taxa in clinically affected ears. This is again mentioned on line 51. As such, please delete either line 46-47 or line 51 – mid 52.

Line 57: add “the” before clinician

Line 63:  correct “at least 10t7” to 9.1 x 107 16S copies per sample

Figure 1 legend: “from here” where? Add an arrow to show where to take the culture or add in ear canal (vertical or horizontal). Also 16s rRNA sequencing; needs to be changed to 16s rDNA sequencing in this legend as they sequenced the DNA in the Tang paper.

Line 75: remove the word “and” before various and place a comma after the reference bracket e.g. [8],

Line 75: place a comma after the word frostbite, remove the word various

Line 80: add a reference for the first sentence starting on line 78.

Line 82: add the phrase “of ETUD” after cases.

Figure 2 legend: Which is the before ear and the after the treatment ear? Please clarify.

Line 88: the phrase typically at a dose of 0.5 mg/kg…once or twice daily? This is not the dose commonly used in practice for dermatologist in our country. We use 0.6 mg/kg PO q.24 h. I think it would be better to add the package label of the therapeutic oral dosage of 0.4 – 0.6 mg/kg

Line 90: add the term “activator cytokines” after the work of and before IL-2

Line 90: after IL-15, add “pro-inflammatory cytokines” interferon-gamma, IL-18 add “and the anti-inflammatory cytokine, IL-10” or you could also use the term “regulatory cytokine” for IL-10.

Line 90: correct IFN-c to read IFN-γ

Line 93: remove “responded well” and replace with more specific “had complete resolution”

Line 93: 22 dogs in the Colombo reference responded with complete resolution between 1-3 months not 23 months as stated in the manuscript submission

Line 93: add within one month after “relapsed” so the clinician can know what to expect as far as time to recurrence

Line 95: if you used “infectious” instead of Leishmania this would also cover bartonellosis

Line 110: for increasing risk along with age add in increasing adult body weight

Line 111: list the breeds in order from highest OR to lowest: Bull terrier, Saint Bernard, French Bulldog etc.

Line 132: I do not know of a reference where myrigotomy is a proposed method to obtain a diagnostic sample for histology in the dog. Please reference this or change histology to cytology which is much more common.

Line 134: fluid removed via suction

Line 142: replace knife with “myringotomy needle” per the paper methods

Line 145: add “the colour of the fluid obtained from the middle ear was judged as either yellow or colourless.

Line 148: place a reference at the end of this sentence

Line 149: remove “the” before the word under

Line 151: remove “the” before ventral and add access to the tympanic membrane…for clarity

Line 158: add after middle ear “as microbiota in the middle ear and the external ear frequently differ from each other…ref Cole et al 1998 JAVMA microbial flora and antimicrobial susceptibility patterns….

Line 181: remove two dogs….replace with three dogs required a second laser treatment. The paper states “A second laser surgery was required in three ears from three dogs and resulted in an excellent outcome in all three ears.

Line 182: add “antimicrobial-saturated” before ear wicks.

Line 184: remove the word “response” and replace with a scientific word used in histology….”hyperplasia”…they general practitioner should know this word and the meaning

Line 192: pseudomonas bacteria is redundant…change to Pseudomonas aeruginosa

Figure 7: I do not think this figure is warranted in the context of representing the significant clinical findings from the manuscript being referenced, as one of their strong conclusions is “a definition of clinical features confirmed to be associated with bacterial biofilm associated infection is needed.”

 How does this image represent a typical gram negative infection with a biofilm? To me as a dermatologist, viewing this image as presented this could also be representative as a yeast otitis externa due to the accumulation of brownish black debris present on the pinna and in the vertical canal. If possible, find an image with suppurative yellow discharge and erosion in the canal, however, I do not think there is good defined clinical signs to say a biofilm is present or absent, therefore I would not put this figure in.

Line 202: add “otic cytological smears” after clinical; remove “and allow clinicians”; remove “bacilli within” (this was not the initial objective of the study) add the word “the” before polysaccharide matrix. So the first objective was just to establish the staining methods to microscopically identify the biofilm and the second objective was to ID the microbial aggregates ….as is written.

Line 208: “the presence of high density” needs to have the word “material” added as this may be confused with the result of high density micro-organisms as the result from the manuscript was as follow” Although the presence of a high density of micro-organisms had strong agreement between investigators, particularly using PAS stain, this feature was not associated with the presence of AAI using either stain (P > 0.05, >0.6 FK; Table 2).

Line 229: Please let the reader know this was a retrospective study. Perhaps start the sentence with This retrospective study tries..

Line 231: Replace “a total of 65 dogs entered the study with Medical records of 65 client–owned dogs were reviewed…as in a retrospective dogs can’t “enter a study”.

Line 265-266: correct the spelling of Malassezia…remove the “L” and italicize

Line 269: correct bracket } to ]

Line 286: replace “at around” with an average dose of 0.8 mg/kg/day

Line 286: phrase no otic treatment were used during this time…not really true..before EFI1 4 dogs received itraconazole and 23 of 82 ears received topical antifungal treatment.

Line 293: remove the extra period

References: see above 

Author Response

I have taken almost every recommendation on board and amended my script and Figure 7.

It was the most complete, comprehensive and helpful review I have ever had

Reviewer 2 Report

This is an excellent paper. The author must be congratulated for an excellent review and summary of the most recently published papers for otology 

Author Response

Cant help but be pleased with this reviewer :-)

Reviewer 3 Report

Dear author, thank you for your article. It is indeed difficult for any clinician to stay updated with the plethora of journals and research. Therefore, review articles or update articles are needed.

Your article is very interesting, and the clinicians will profit after reading it but in my opinion there must be done some changes before considering acceptance for publication. See below my comments:

Line 24: Very interesting topic! This paragraph is very important nowadays. I recommend doing the following changes. 16S new technology as a title is not very informative. Please consider 16S rRNA gene sequencing. Also, 16S is used only for bacteria. Therefore, you might want to change the word microflora to bacterial microbiota.

Lines 25-29: you should add more specific details about the 16S sequencing methodologies that are used nowadays (e.g. Illumina amplicon, pyrosequensing etc). That is important for the readers to be informed what different 16S technologies are used and also because we should avoid comparing direct 1:1 studies using different 16S technologies. Or at least a methodology bias should be mentioned when comparing.

Lines 30-35: 16S rRNA is used for analysis up to genus level. Not species. The results you present here are Shotgun-based sequencing results. The correct terminology should be used. Also if you decide to present also Shotgun-seq  results then you must change the title of the paragraph (is not a16S technology any more).

Line 32-33 : what diversity was different? Alpha or beta? Please specify. Also I cannot fully understand which group had the 30 different species (the normal or the otitis group). Did one group only had 30 species? The other had less?

Line 34: the word isolates is not appropriate when used for sequencing methods.

Figure 1: yeasts are referred here: 16S cannot be used for yeast seq. That can cause some  misunderstanding.

Line 44-53: I could not understand if these findings are based on culture dependent or culture independent studies. Please specify. Also add reference(s).

Line 68: please add a conclusion answering your questions in line 57.

Section 2 in general: As above said is an important topic. However only one study has been cited, when the last 4 years several couple more studies have been published in many different journals. I think a review paper should summarize the findings of all those studies, since this is a new technology. Some studies examine only the ears and some the skin and the ear. I recommend you add the findings of all those studies (only the results form the ears if skin swabs are also used, since the title of the current article is otology), which the purpose of a review article.

Section 3:

Line 74 -75: The way that the differentials are presented could be constructed in a more “non-dermatologists” friendly way. I suggest you construct your differentials as inflammatory infectious, inflammatory noninfectious etc… and add the specific diseases to the correct category.

Line 78: You write that ETUD is usually an idiopathic diagnosis? I am confused as you state above infectious causes. Proliferative thrombovascular necrosis of the pinnae is usually idiopathic. Also make a connection with oclacitinib and the category of causes we target with oclacitinib. Otherwise, someone might think we should treat with oclacitinib a ETUD due to Bartonella.

Lines 78-81: The histopathology you describe here is the pathology of the proliferative thrombovascular necrosis of the pinnae. ETUD might have different pathology based on the cause. Please change that or delete it. Is the pathology adding any value to your manuscript?

Lines 87-90: please add here what doses have been used in the studies you cite. And also please make clear for the reader which dose causes which cytokine change.

Lines 90-99: please add the initial dose the dogs where treated as well as the long term required dose.

Section 4

Line 109: please make clearer that this is the new study that delivers new findings.

Lines 119-129: It is not clear to me if based on this study these dogs with these changes can get aural hematoma without another concurrent reasons (e.g. head shaking due to otitis etc.). Can you please add some language about that?

Section 6

Line 179: Why was CsA included to the post-operative treatment. There should be a reason? And also, I am wondering if CsA can cause wound healing problems. I think a review should have comment on that based on the literature (if the authors of the study have not).

Line 190: please add that further studies should evaluate the CO2 Laser surgery of this entity vs cold steel surgery. This paper was a case series and the results should be carefully used.

Section 9: the paper you cite 26 is aa retrospective study. Please add that information. After reading the original study I think some important findings or and comments should be addressed here too. The overall message I got after reading this section was not the same after reading the original study.

Thank you again for your manuscript. Pending the above mentioned changes, I believe a revised version of the manuscript could be suitable for publication.

Author Response

Almost all his points have been addressed in my response to the other review

32-33 I think starting to introduce alpha and beta diversity will mean having to explain the terms in addition to mentioning them. I think too technical for the average reader

Figure 1. line amended to specify only bacteria isolated 

44-53, personal observation added

Section 2. this is not a review of oclacitinib and its uses in dermatology and ototology (although I will gladly write one) but a summary of a paper which describes it use

78-81 the author states idiopathic and describes the histopathology not me.

I think the other points are already taken on board

Reviewer 4 Report

Interesting and useful article for the clinical veterinarian. In Figure 1 a hand with gloves would be better ...

The main question addressed by the research is the importance of updating some clinical and therapeutic innovations in the veterinary otology field.

I consider the original topic in otology because it deals with useful and innovative novelties for veterinary clinical practice.

The article allows you to update on previous works published quickly.

The conclusions are  consistent with the evidence and arguments presented, and do  address the main question.

The references appropriate.

Author Response

I did not wear gloves when the picture was taken

No ther points to take on board

Round 2

Reviewer 1 Report

Thank you for addressing the reviewers comments. I really like how this paper reads now. Just one aspect needs to be addressed. The added line 199-200 needs to be revised. The word "lasern" and then the last part of the sentence does not seem to be complete.

Thank you for contributing a dermatology manuscript for the general practitioner as these current summaries are not only needed but often read by many.

Author Response

Accept all amendments

Reviewer 3 Report

Great work.

Author Response

Reviewer’s points noted